# Reinforcement Learning Enhanced Explainer for Graph Neural Networks

**Caihua Shan**[1], **Yifei Shen**[2], **Yao Zhang**[3], **Xiang Li**[4], **Dongsheng Li**[1]
[1]Microsoft Research Asia
`{caihua.shan,dongsheng.li}@microsoft.com`
[2]The Hong Kong University of Science and Technology
`yshenaw@connect.ust.hk`
[3]Fudan University
`yaozhang@fudan.edu.cn`
[4]East China Normal University
`xiangli@dase.ecnu.edu.cn`

## Abstract

Graph neural networks (GNNs) have recently emerged as revolutionary technologies for machine learning tasks on graphs. In GNNs, the graph structure is generally incorporated with node representation via the message passing scheme, making the explanation much more challenging. Given a trained GNN model, a GNN explainer aims to identify a most influential subgraph to interpret the prediction of an instance (e.g., a node or a graph), which is essentially a combinatorial optimization problem over graph. The existing works solve this problem by continuous relaxation or search-based heuristics. But they suffer from key issues such as violation of message passing and hand-crafted heuristics, leading to inferior interpretability. To address these issues, we propose a RL-enhanced GNN explainer, *RG-Explainer*, which consists of three main components: starting point selection, iterative graph generation and stopping criteria learning. RG-Explainer could construct a connected explanatory subgraph by sequentially adding nodes from the boundary of the current generated graph, which is consistent with the message passing scheme. Further, we design an effective seed locator to select the starting point, and learn stopping criteria to generate superior explanations. Extensive experiments on both synthetic and real datasets show that RG-Explainer outperforms state-of-the-art GNN explainers. Moreover, RG-Explainer can be applied in the inductive setting, demonstrating its better generalization ability.

## 1   Introduction

Graph Neural Networks (GNNs) extend neural network models on ubiquitous graph data via utilizing the message passing scheme to incorporate graph structures with node features. They have achieved state-of-the-art performance not only in classic machine learning tasks on graphs, e.g., node classification [10, 28], link prediction [38], and graph classification [33], but also in reasoning tasks, e.g., intuitive physics [4], mathematical reasoning [24], and IQ tests [3]. Similar to most deep learning methods, one major limitation of GNNs is the lack of the interpretability for the predicted results; a post-hoc analysis is usually needed to explain the results.

To enhance the interpretability of GNNs, a line of works [34, 17, 30, 37, 31] focused on developing GNN explainers. The goal of GNN explainers is to identify a most influential subgraph structure to interpret the predicted label of an instance (e.g., a node or a graph). It can be generally formulated as

35th Conference on Neural Information Processing Systems (NeurIPS 2021).

an optimization problem that maximizes the mutual information between the predicted results and the distribution of relevant subgraphs under some size constraints.

The pioneering works, e.g., GNNExplainer [34] and PGExplainer [17], attempt to solve the optimization problem with continuous relaxation. These methods optimize a soft mask matrix for edges, and select the important nodes/edges by the threshold. However, they cannot guarantee that nodes and edges in the output subgraph are connected. Thus, their explanatory subgraphs cannot explicitly visualize the message passing paths. Besides, they consider the importance of each edge independently, and ignore the interactions among selected nodes and edges. Some recent works, such as SubgraphX [37] and Causal Screening [31], design the search criteria and use search-based methods to solve the optimization problem. Due to the combinatorial property of searching explanatory graph structures, it is difficult to design a general hand-crafted search criterion. These criteria are limited on specific situations and thus not widely applicable.

To address these issues, we propose RG-Explainer, which adopts reinforcement learning to explain GNNs' predictions. Our framework is inspired by classic combinatorial optimization solvers, which consists of three crucial steps: *starting point selection*, *iterative graph generation* and *stopping criteria learning*. These three components work together to generate an explanatory graph that interprets the predicted label of a given node/graph instance, as we elaborate next.

Firstly, starting point selection selects the most important node as the seed node in the instance. If the task is to interpret the prediction of a specific node label, then the most important node refers to the node itself. To explain a graph label, we design a seed locator to learn the node that influences the graph label the most. Iterative graph generation is the key module in our method, which generates the nodes in the explanatory graph sequentially. Specifically, we add an influential node (action) from the neighbors based on the current generated graph (state) at each step. It explicitly guarantees the connectivity of the generated graph. The generation process is controlled by the reward, i.e., the mutual information between the original predicted label and the label made by the generated graph. To ensure a compact and meaningful explanatory graph, we also involve some constraints into the reward, such as size loss, radius penalty and similarity loss. Finally, stopping criteria are learned to further avoid generating very large explanatory graphs.

Furthermore, our method has better generalization ability and can be applied in both transductive and inductive setting. Different from the search-based methods, we learn the heuristics from the data automatically. A well-trained RG-Explainer can infer the explanations of instances which are not involved in the training phase.

We conduct extensive experiments on both synthetic and real-world datasets to show that the proposed RG-Explainer can achieve superior performance compared to state-of-the-art GNN explainers. In particular, our visualization results further demonstrate the better intepretability of our method.

## 2  Related Work

**Graph Neural Networks.**    Graph neural networks (GNNs) have achieved great success on graph-structured data in many real-world applications, including recommender systems, chemistry and bioinformatics [8, 11, 26]. The majority of GNNs used today follow the message passing scheme [8], which aggregate information from neighbors with different aggregation functions, like mean/max/LSTM-pooling in GCN [13] and GraphSAGE [10], sum-pooling in GIN [33], attention mechanisms in GAT [28], etc. SGC [32] observes that the superior performance of GNNs is mainly due to the neighbor aggregation rather than feature transformation and nonlinearity, and proposed a simple and fast GNN model. APPNP [14] shares the similar idea by decoupling feature transformation and neighbor aggregation.

**Graph Generation.**    There have been a variety of methods for graph generation. RVAE [20] is a variational auto-encoder (VAE) based method with a regularizer to ensure semantic validity. Normalizing flow based methods including GraphNVP [21], GraphAF [27], and GraphDF [18] utilize invertible neural networks to define mappings between latent variables and data points. Generative adversarial networks (GANs) [9] based methods like MolGAN [7] and GCPN [35] involve a generator and a discriminator, where the generator is adversarially trained to fool the discriminator.

From the perspective of graph generation process, they can be classified into one-shot generation and iterative generation. RVAE and MolGAN directly generate adjacency matrices, while GraphAF, GraphDF and GCPN generate graphs by sequentially adding new nodes and edges. Though our proposed RG-Explainer is an iterative generation method, RG-Explainer is different from the graph generation methods in that the above methods generate graph out of the air, while RG-Explainer needs to dynamically select suitable subgraphs to explain the predictions.

**Graph Combinatorial Optimization with RL.** With the success of deep reinforcement learning in games [22], researchers have attempted to utilize RL techniques for the graph combinatorial optimization problems [6, 19, 39, 25]. Specifically, S2V-DQN [6] uses deep Q-learning with graph embedding to learn effective algorithms for the Minimum Vertex Cover, the Maximum Cut and the Traveling Salesman problems. A graph pointer network is proposed in [19] to solve the TSP efficiently. Further, Seal [39] learns heuristics to detect communities in the graph with policy gradient. Note that explaining GNNs is also a combinatorial optimization problem. Thus, in this paper, we propose a RL-based framework with three dedicated steps to generate explanations.

**Post-hoc Analysis in Graph Neural Networks.** By extending existing image/text explanation techniques to the graph, some gradient-based methods [23, 1] are proposed to study the importance of nodes and edges in the graph. However, their performances have been proved to be sub-optimal [34] because they cannot incorporate the special properties of graphs.

GNNExplainer [34] is the first specific method proposed to explain trained GNNs. It defines the problem as an optimization task, which maximizes the mutual information between the predicted labels and the distribution of possible subgraphs under some constraints. Following the problem setting, PGExplainer [17] leverages the representations generated by the trained GNN and adopts a deep neural network to learn the crucial nodes/edges. These methods both utilize the continuous relaxation on edges, and add size and entropy constraints to make the explanation small and sparse. Specifically, they optimize a soft mask matrix for edges, and select crucial nodes/edges by the threshold. However, they compute the importance of each edge independently, which may lead to a disconnected explanatory graph with information redundancies. Our model sequentially adds important nodes from the neighbors of the current generated graph, which considers the information already involved in the current graph and ensures the connectivity.

SubgraphX [37] uses Monte Carlo tree search and Shapley value as a score function to find the best connected subgraphs as explanations for GNNs. Causal Screening [31] is another search-based method, but it uses greedy search and causality measure to generate the explanations. Different from the search-based methods where heuristics are usually hand-crafted, our method uses RL to learn heuristics from data, which can be widely applicable. Besides, our learning-based method could train by a small set of instances, and infer the explanations of many other similar unseen instances much faster than the search-based methods.

Different from the instance-level explanation, there also exists the model-level explanation to investigate general patterns for predictions. For example, XGNN [36] utilizes the graph generator to interpret GNNs at the model-level. In particular, the instance-level explainer interprets the prediction for a certain given instance while the model-level explainer is input-independent and less precise.

## 3 Preliminary

In this section, we first introduce the notations used and then give the formal problem definition.

**Graph.** Let $G = (\mathcal{V}, \mathcal{E})$ denote the graph with node set $\mathcal{V} = \{v_1, v_2 \cdots v_N\}$ and edge set $\mathcal{E} \in \mathcal{V} \times \mathcal{V}$. Nodes in $\mathcal{V}$ could be associated with $d$-dimensional node features $\mathcal{X} \in \mathbb{R}^{N \times d}$. The graph $G$ is described by the adjacency matrix $A$ such that each entry $A_{ij} = 1$ if $e_{ij} \in \mathcal{E}$; 0, otherwise. $\widetilde{A} = A + I_N$ denotes the adjacency matrix with added self-loops. A symmetrically normalized adjacency matrix with self-loops $\widehat{A}$ could be computed by $\widetilde{D}^{-1/2} \widetilde{A} \widetilde{D}^{-1/2}$, where $\widetilde{D}$ is the diagonal degree matrix of $\widetilde{A}$.

**Message Passing.** Given an input graph $G$ and node features $\mathcal{X}$, a GNN model $f(G, \mathcal{X})$ learns node representations. To fuse the information of both node features and graph topology in node

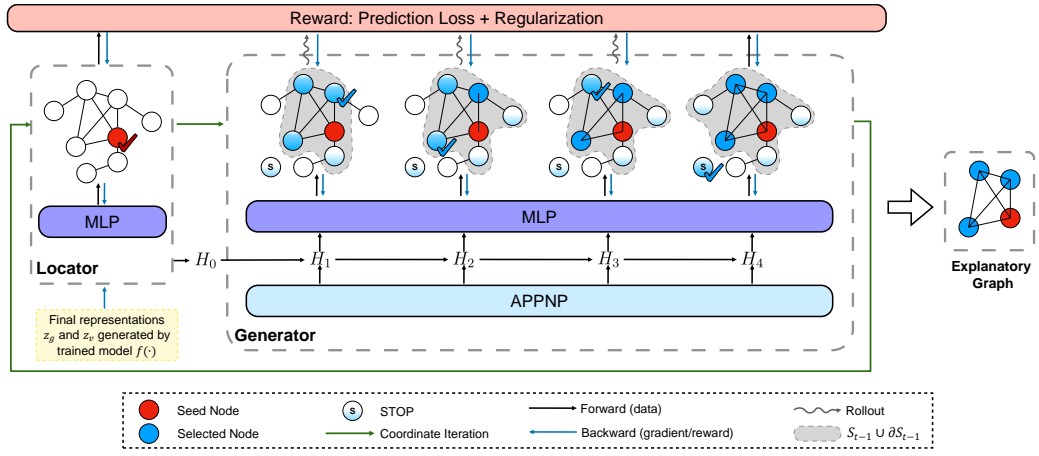

Figure 1: Illustration for explaining GNNs on graph classification. It contains two main components: a locator and a generator. The locator selects a seed node and feeds it into the generator. Based on the seed node, the generator then generates a complete explanatory graph and obtains the final reward from the environment. This reward is used as feedback to update the generator and the locator. We alternatively optimize the generator and locator until convergence.

representation vectors, GNN models utilize the message passing scheme to aggregate information from node neighbors. At each layer $l$, a node $v$ aggregates all the messages in the $(l-1)$-th layer from all its neighbors to generate its embedding: $h_v^l = \textbf{update}(\sum_{w \in N(v)} \textbf{message}(h_v^{l-1}, h_w^{l-1}), h_v^{l-1})$, where $N(v)$ is the neighbor set of node $v$. With $L$ layers, each node $v$ generates its embedding vector $z_v = h_v^L$ from the $L$-hop neighborhood. In the node classification task, we train an classifier with $z_v$ as input to predict the label for node $v$. In the graph classification task, we first use an aggregation function $\textbf{readout}(\{z_v\})$ to generate the graph representation, which is further fed into a classifier to predict the label.

**Problem Definition.** Given an input graph $G = (\mathcal{V}, \mathcal{E})$ and a trained GNN model $f(\cdot)$, GNN explainers aim to generate an explanatory subgraph $S = (\mathcal{V}_S, \mathcal{E}_S)$, where $\mathcal{V}_S \in \mathcal{V}$ and $\mathcal{E}_S \in \mathcal{E}$. The goal is to maximize the mutual information between the original label prediction $Y = f(G)$ and the label prediction distribution based on the generated explanatory subgraph. Formally, the objective can be given as $\max_S \text{MI}(Y, S) = H(Y) - H(Y|S)$, where $MI(\cdot)$ is the mutual information function and $H(\cdot)$ is the entropy function. Since $H(Y)$ is fixed in the explanation stage, the objective can be rewritten as $\min_S H(Y|S)$.

# 4 Methodology

The objective $\min_S H(Y|S)$ is intractable since there are exponential candidates for $S$. It can be considered as a combinatorial optimization problem, where we need to choose a subset of nodes in the graph to optimize the objective. We explore how RL can be used to iteratively understand the representations produced by GNNs, and generate the explanatory subgraph optimizing the objective. Three proposed components will be described in the following.

## 4.1 Iterative Graph Generation (Graph Generator)

Given a starting point $v_0$, the graph generator is used to generate a connected subgraph $S = \{v_0, v_1 \cdots v_T\}$, where we select one node in a step and $T$ is the total number of steps. Specifically, at the $t$-th step, we have the current partial solution $S_t = \{v_0, v_1 \cdots v_{t-1}\}$. We next select a new node $v_t$ from the boundary $\partial S_{t-1}$ and expand the solution $S_t = S_{t-1} \cup \{v_t\}$. The state is defined as the combined representation of both $v_0$ and $S_{t-1}$. The action space is the boundary $\partial S_{t-1} = \cup_{v \in S_{t-1}} N(v) \backslash S_{t-1}$. We further associate the solution with a reward value.

**State.** At the $t$-th step, we first augment each node feature vector by adding the information of the starting point and the current partial subgraph. For each node $v$ in $S_{t-1}$ or $\partial S_{t-1}$, we concatenate two values with its original feature vector $x_v$:

$$x'_v = [x_v, \mathbb{1}_{\{v \in \{v_0\}\}}, \mathbb{1}_{\{v \in S_{t-1}\}}], \quad X'_t = [x'_v]_{\forall v \in S_{t-1} \cup \partial S_{t-1}}, \tag{1}$$

where $\mathbb{1}$ is the indicator function and $\mathbb{1}_{\{v \in S\}} = 1$ if $v \in S$ otherwise 0. We concatenate the augmented node features and obtain the initial state representation $X'_t$.

Each node could further combine information from its current neighborhood. To achieve this, we utilize some existing GNN methods, e.g., APPNP [14], which separate the non-linear transformation and information propagation. These methods have shown to be highly efficient and also effective. Specifically, we have the following update equation:

$$H_t^{(0)} = \Theta_1 X'_t, \quad H_t^{(l+1)} = (1 - \alpha)\widehat{A}H_t^{(l)} + \alpha H_t^{(0)}, \tag{2}$$

where $\Theta_1$ is the trainable weight matrix, $\widehat{A}$ is the symmetrically adjacency matrix, and $\alpha$ is a hyper-parameter used to control weight. After $L$-layer updates, we obtain the node representations $H_t^L$. we feed them into a MLP to improve the representation ability:

$$\bar{H}_t(v) = \text{MLP}(H_t^L(v); \Theta_2), v \in S_{t-1} \cup \partial S_{t-1}, \tag{3}$$

where $\Theta_2$ is the learnable parameters in the MLP. At the $t$-th step, we only consider the nodes in $S_{t-1}$ and $\partial S_{t-1}$ because other nodes do not influence the state and action space.

**Action.** Since the connectivity of the generated subgraph is required, we take $\partial S_{t-1}$ as the action space at the $t$-th step. We utilize a softmax function to calculate the probability of taking an action $v \in \partial S_{t-1}$:

$$a_t(v) = \frac{\exp(\theta_3^T \bar{H}_t(v))}{\sum_{u \in \partial S_{t-1}} \exp(\theta_3^T \bar{H}_t(u))}, \ v \in \partial S_{t-1}, \tag{4}$$

where $\theta_3$ is the trainable parameter vector.

**Objective.** Following [17], we use the cross-entropy function to replace the conditional entropy function $\min_S H(Y|S)$ with $N$ given instances. We rewrite the objective as:

$$\text{Prediction Loss} = -\frac{1}{N} \sum_{n=1}^{N} \sum_{c=1}^{C} P(Y = c) \log P(f(S_n) = c), \tag{5}$$

where $S_n$ is the explanatory subgraph for the $n$-th instance, $C$ is the number of possible predicted labels, $P(Y = c)$ is the probability that the original output of the trained GNN $f$ is $c$, and $P(f(S_n) = c)$ is the probability that the label prediction of $f$ on the subgraph $S_n$ is $c$.

We further introduce some regularization terms to restrict the characteristics of the explanatory subgraph. To obtain a compact and succinct explanatory subgraph $S$, we define a size loss and a radius penalty, respectively. The size loss is used to limit the number of nodes in $S$ while the radius penalty can compute the longest length of the shortest path from the seed node to other nodes in $S$. We also introduce a similarity loss that measures the similarity between the original node representation $z_{v_0}$ and the new representation generated on $S$. Formally, these loss function are defined as:

$$\text{Size Loss} = ||S||_1, \quad \text{Radius Penalty} = \max_{u \in S} \text{Distance}(v_0, u), \quad \text{Similarity Loss} = ||\bar{H}_T(v_0) - z_{v_0}||_2. \tag{6}$$

The final objective is to minimize

$$\mathcal{L}(S) = \text{Prediction Loss} + \lambda_1 \cdot \text{Size Loss} + \lambda_2 \cdot \text{Radius Penalty} + \lambda_3 \cdot \text{Similarity Loss}, \tag{7}$$

where $\lambda_*$ are hyper-parameters to control the term importance.

**Reward.** We take the objective function loss $\mathcal{L}$ as the (negative) reward. However, the loss cannot be separated into each generation step. If we simply compute $-\mathcal{L}(S_t)$ at the $t$-th step and regard it as the reward $r_t$ for the state-action pair $(s_t, a_t)$, it could lead to the sub-optimal results. Therefore, we do not compute intermediate rewards when adding a new node to the subgraph. We only return the reward $-\mathcal{L}(S)$ when we complete the generation process of $S$.

**Optimization with policy gradient.** We learn the graph generator $\mathbb{G} = \{\Theta_1, \Theta_2, \theta_3\}$ via policy gradient. The policy $\pi_\theta$ is learned to maximize $\mathbb{E}_{S|v_0 \sim \mathbb{G}}[-\mathcal{L}(S)]$, whose policy gradient is

$$\nabla \mathbb{E}_{S|v_0 \sim \mathbb{G}}[-\mathcal{L}(S)] = \mathbb{E}_{v_1,\ldots,v_T|v_0 \sim \mathbb{G}} \left[ \sum_{t=1}^{T} \nabla \log \pi_\theta(v_t|S_{t-1}) \cdot Q(S_{t-1}, v_t) \right]. \tag{8}$$

Here, $S_0$ is the given seed $v_0$ and $Q(S_{t-1}, v_t) = \mathbb{E}_{v_{t+1},\ldots v_T|S_t \sim \mathbb{G}}[-\mathcal{L}(S)]$ is the state-action value function. Specifically, we use the Monte-Carlo estimation to approximate $Q$ values:

$$Q(S_{t-1}, v_t) = \begin{cases} \frac{1}{M} \sum_{i=1}^{M} -\mathcal{L}(S^{(i)}) & \text{when} \quad t < T \\ -\mathcal{L}(S_{t-1} \cup v_t) & \text{when} \quad t = T \end{cases} \tag{9}$$

where $S^{(i)} (i = 1, \ldots, M)$ are rollouts (i.e., complete explanatory graphs) sampled from the policy given the partial solution $S_{t-1}$ and $v_t$. In this way, we can distribute reward signals at all steps.

## 4.2 Stopping Criteria Learning

We further learn the stopping criteria to judge the goodness of the current generated graph. We add a special STOP action into the action space to learn node selection and stopping criteria simultaneously. Since we have already obtained the node representation $\bar{H}_t$ in the current state, the STOP action could aggregate the representations with self-attention mechanism:

$$\gamma_t(v) = \frac{\exp(\theta_4^T \bar{H}_t(v))}{\sum_{u \in \{S_{t-1} \cup \partial S_{t-1}\}} \exp(\theta_4^T \bar{H}_t(u))}, \quad \bar{H}_t(\text{STOP}) = \sum_{v \in \{S_{t-1} \cup \partial S_{t-1}\}} \gamma_t(v) \bar{H}_t(v). \tag{10}$$

where the parameter $\theta_4$ helps to learn the attention $\gamma_t(v)$ for each node $v$ in the current state.

After having $\bar{H}_t(\text{STOP})$, we could put it into Eqn. 4, and compute $a_t(\text{STOP})$ and $a_t(v)_{\forall v \in \partial S_{t-1}}$ together. In practice, we also set a maximum number of generation steps to avoid generating very large subgraphs.

## 4.3 Starting Point Selection (Seed Locator)

For node classification tasks, the starting point is the node instance whose predicted label needs to be interpreted. However, the starting point is difficult to select for a graph instance. To solve the problem, we need to construct a seed locator $\mathbb{L}$ to first identify the most influential node in the graph and then generate the explanatory subgraph from that node.

Given $N$ graph instances $g_n$, the objective in Eqn. 5 could be rewritten with the locator $\mathbb{L}$:

$$\min_{\mathbb{G}, \mathbb{L}} -\frac{1}{N} \sum_{n=1}^{N} \sum_{c=1}^{C} P(f(g_n) = c) \log P(f(\mathbb{G}(\mathbb{L}(g_n))) = c), \tag{11}$$

where both the generator $\mathbb{G}$ and the locator $\mathbb{L}$ are to be learned. The regularization terms in Eqn. 6 can be further added to the objective for more constraints. We train $\mathbb{G}$ and $\mathbb{L}$ coordinately, i.e., we fix the parameters in one module and train the other module to optimize the objective in Eqn. 11 iteratively.

When we fix $\mathbb{L}$, the way to train $\mathbb{G}$ is the same as described in Sec. 4.1. Here we introduce how to construct $\mathbb{L}$ when $\mathbb{G}$ is fixed. Based on $\mathbb{G}$, we can generate a subgraph $S$ for each node in the graph instance and compute the corresponding reward $-\mathcal{L}(S)$. A straightforward way is to enumerate over all the nodes and select the node with the highest reward. However, such a brute-force method is computationally infeasible when we have many graph instances. Therefore, we adopt a learning-based method. Specifically, we use a three-layer MLP to model the influence of a node $v_{i,n}$ on the label of the graph instance $g_n$:

$$\omega_{i,n} = \text{MLP}([z_{g_n}, z_{v_{i,n}}]), \tag{12}$$

where $z_{g_n}$ and $z_{v_{i,n}}$ are the final representations of the graph instance $g_n$ and the node $v_{i,n}$ generated by the trained model $f(\cdot)$, respectively. Because the goal of $\mathbb{L}$ is to return the node $v_{i,n}$ with highest $\omega_{i,n}$ for graph $g_n$, we utilize the Kullback-Leibler divergence loss, KLDivLoss($\omega_{i,n}, -\mathcal{L}(\mathbb{G}(v_{i,n}))$), which aims to make the distribution between estimated values $\omega_{i,n}$ and the actual reward of explanatory subgraph produced by the current generator close. The softmax layers are used to transform $\omega_{i,n}$ and $-\mathcal{L}(\mathbb{G}(v_{i,n}))$ into two distributions over the nodes $v_{i,n}$ in graph $g_n$. We sample graph instances to train the MLP, and let $\mathbb{L}$ learn what kind of seed nodes has the highest reward to minimize Eqn.11.

### 4.4 Pre-training

In this section, we show the pre-training strategies with Maximum Log-Likelihood Estimation (MLE) that can be used to initialize the generator $\mathbb{G}$ and the locator $\mathbb{L}$, respectively.

**Pretrain $\mathbb{G}$.** The generator $\mathbb{G}$ produces an unordered set $S$ but in an ordered sequence. We maximize over all possible generated orderings for an explanatory graph [29]:

$$\max_{\tau} \sum_{i=1}^{T} \log \mathbb{G}(v_{\tau(i)} | \{v_0, v_{\tau(1)}, ..., v_{\tau(i-1)}\}), \tag{13}$$

where $\tau$ is any *valid* permutation given $S$. Here, the validity means that each $v_{\tau(i)}$ should be in the boundary of $\{v_0, v_{\tau(1)}, ..., v_{\tau(i-1)}\}$.

Considering there are at most $T!$ orderings in the worst case, we use a bootstrapped way with set2set [29] to approximate Eqn. 13. Specifically, we optimize Eqn. 13 by maximizing the following set-wise log-likelihood instead:

$$\sum_{i=1}^{T} \log \mathbb{G}(v_{\tau^*(i)} | \{v_0, v_{\tau^*(1)}, ..., v_{\tau^*(i-1)}\}), \tag{14}$$

where $\tau^*(i) = \arg\max \mathbb{G}(\cdot | \{v_0, v_{\tau^*(1)}, ..., v_{\tau^*(i-1)}\})$. Here we utilize the 3-hop neighborhood of seed nodes as the initial explanations $S$ (i.e., pre-training samples). After constructing the samples, we train $\mathbb{G}$ to maximize the set-wise log-likelihood in Eqn. 14. Vinyals et al. [29] also pointed out if we naively optimize it, the model would pick a random ordering and get stuck on it. Thus, a list-wise log-likelihood is also necessary to explore the space of ordering. For each pretrain sample $S$, we choose a valid permutation $\tau'$ with random lengths in advance, and optimize the list-wise MLE: $\sum_{i=1}^{T} \log \mathbb{G}(v_{\tau'(i)} | \{v_0, v_{\tau'(1)}, ..., v_{\tau'(i-1)}\})$.

**Pretrain $\mathbb{L}$.** We pretrain the locator $\mathbb{L}$ without the generator $\mathbb{G}$. Similar as in $\mathbb{G}$, we utilize the 3-hop neighborhood of a node as the initial explanatory subgraph $S$. We randomly sample some nodes in the graph instances and compute the rewards of their 3-hop neighborhoods. These samples are used to pretrain the parameters in $\mathbb{L}$.

## 5 Experiments

In this section, we first introduce our experimental setup. Then we compare RG-Explainer with two state-of-the-art baselines GNNExplainer [34] and PGExplainer [17] in both qualitative and quantitative evaluations. Further, we evaluate the performance of our method in the inductive setting. Due to the space limitation, we move the pseudocode, implementation details and ablation study to the supplementary materials. We also attach our codes in the supplementary materials.

### 5.1 Setup

For fairness, we follow the experimental setup in [17, 12], i.e., the same datasets, trained GNN model and evaluation metrics. Besides, we also utilize the same fine-tuned parameters in [12] for our competitors, GNNExplainer and PGExplainer.

**Datasets.** We use six datasets, in which four synthetic datasets (BA-shapes, BA-Community, Tree-Cycles and Tree-Grid) are used for the node classification task and two datasets (BA-2motifs and Mutagenicity) are used for the graph classificition task. These datasets are composed of *motifs* and *bases*. The motif is a small but important substructure in a graph, which has been shown to play a crucial role in predicting the label of node/graph instances [5, 15, 16]. The base is the remaining part of a graph which is randomly generated. Motifs are taken as the ground-truth and the goal of explainers is to find them. Details of these datasets are described as follows.

(a) The BA-shapes dataset consists of one Barabasi-Albert(BA) graph [2] as the base and 80 house-structure motifs. Each motif is randomly attached to a node in BA graph and extra edges are added as noises. (b) The BA-community dataset is comprised of two BA-shapes with different node features generated by Gaussian distributions. The extra edges are also added to connect two BA-shapes. (c)

Table 1: Visualization (Qualitative Evaluation)

| | Node Classification | | | | Graph Classification | |
|---|---|---|---|---|---|---|
| | BA-Shapes | BA-Community | Tree-Cycles | Tree-Grid | BA-2motifs | MUTAG |
| Explanations by GNN-Explainer |  |  |  |  |  |  |
| Explanations by PG-Explainer |  |  |  |  |  |  |
| Explanations by RG-Explainer (ours) |  |  |  |  |  |  |
| Ground-Truth Motif |  |  |  |  |  |  |

The Tree-cycles dataset includes a multi-level binary tree as the base and 80 six-node cycle motifs. The cycle motifs are randomly attached to the tree. (d) The Tree-grid dataset is similar to Tree-cycles, which uses the $3 \times 3$ grid motifs instead. (e) The BA-motifs dataset has 1000 graphs where half of them are a BA graph attached with a house-structure motif, while the rest are a BA graph attached with a five-node cycle motif. (f) The Mutagenicity dataset is a real dataset, which includes 4337 molecule graphs. They can be classified as mutagenic or nonmutagenic depending on whether having $NH_2$ or $NO_2$ motifs.

**Model.** We use the trained GNN model in [12], whose architecture is given in [17, 34]. Specifically, the model that consists of three consecutive Graph Convolution layers connected with a fully connected layer is used for node classification. For graph classification, the model includes three consecutive Graph Convolution layers fed into two max and mean pooling layers, respectively. The two pooling layer output embeddings are then concatenated to generate the input for a fully connected layer for graph classification.

**Metrics.** The motifs in each dataset are the ground-truth explanations. The edges in the motif are positive and other edges are negative. GNNExplainer and PGExplainer return a mask matrix to represent the importance of each edge in the instance. Our method generates a subgraph. Based on the generation order of edges, we could also assign different weights to edges. Therefore, the explanation problem can be formalized as a binary classification task, where edges in the ground-truth motif are taken as prediction labels and the weights of edges are viewed as prediction scores. With the explanatory subgraph provided by explainers, the **AUC score** can be computed to measure the accuracy for quantitative evaluation.

## 5.2 Qualitative evaluation

Table 1 visualizes some examples of explanatory graphs on all the datasets. For node classification, we amplify the center node instance and generate the subgraph from it. For graph classification, the graph represents the whole graph instance. We use different colors to denote node labels.

Intuitively, a superior explainer should include more edges in the ground-truth motif and less irrelevant nodes and edges in the explanatory subgraph. To show whether explainers assign larger weights to edges in the motif, we use the bold black edges to represent the top-$k$ edges, where $k$ is the number of edges inside the ground-truth motif.

For easy cases, all the methods could find the ground-truth motif. Thus, we choose some difficult instances to show advantages of our method. From the table, we see that all the three methods can generate subgraphs that contain the ground-truth "house" motif on both BA-Shapes and BA-

Table 2: Explanation AUC (Quantitative Evaluation).

| | Node Classification | | | | Graph Classification | |
|---|---|---|---|---|---|---|
| | BA-Shapes | BA-Community | Tree-Cycles | Tree-Grid | BA-2motifs | MUTAG |
| GNNExplainer | $0.742 \pm 0.006$ | $0.708 \pm 0.004$ | $0.540 \pm 0.017$ | $0.714 \pm 0.002$ | $0.499 \pm 0.004$ | $0.606 \pm 0.003$ |
| PGExplainer | $\mathbf{0.999 \pm 0.000}$ | $0.825 \pm 0.040$ | $0.760 \pm 0.014$ | $0.679 \pm 0.008$ | $0.133 \pm 0.046$ | $0.847 \pm 0.081$ |
| RG-Explainer (ours) | $0.985 \pm 0.013$ | $\mathbf{0.919 \pm 0.017}$ | $\mathbf{0.787 \pm 0.099}$ | $\mathbf{0.927 \pm 0.031}$ | $\mathbf{0.657 \pm 0.107}$ | $\mathbf{0.873 \pm 0.028}$ |
| Improve | -1.5% | 11.4% | 3.6% | 29.8% | 31.7% | 2.8% |

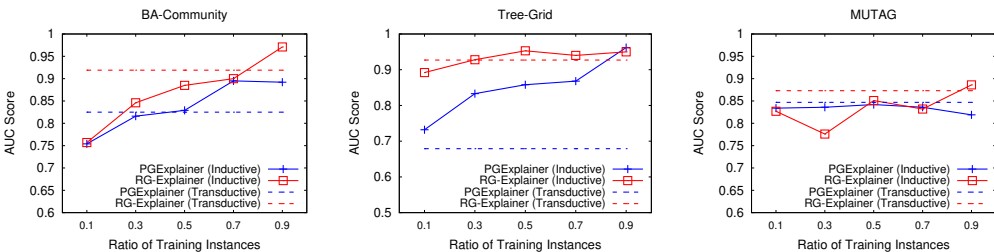

Figure 2: Comparison between RG-Explainer and PGExplainer in the inductive setting.

Community datasets. However, both GNNExplainer and PGExplainer include many irrelevant nodes and edges, while our method adopts the stopping criteria to generate more concise subgraphs. On both Tree-Cycles and Tree-Grid datasets, since we select the connection node between the base and the motif as the node instance, it is hard to identify the ground-truth motif (cycle or grid) exactly. This is because the explanatory subgraphs could easily include the base. Compared with competitors, our method can generate the explanatory subgraphs that contain the complete ground-truth motifs, i.e., edges in the motifs are all marked as black.

For graph classification, explainers should identify the ground-truth motif that decides the label of graph instances. For the instance in BA-2motifs, the ground-truth "house" motif is located at the right bottom. Both GNNExplainer and PGExplainer involve accurate and wrong edges in the explanations, while our method only adds edges in the ground-truth "house" motif to the generated subgraph. For the MUTAG dataset, both PGExplainer and our method identify the correct motif. In particular, our method correctly locates the $O$ atom and obtains the ground-truth motif $NO_2$.

## 5.3   Quantitative evaluation

We next show the quantitative results in Table 2. For each method, we compute the average AUC scores and standard deviations over 10 runs. Note that the AUC scores reported here are a bit different from that in the original papers due to the unstable convergences [12].

From the table, we see that our method RG-Explainer achieves the best results on 5 out of 6 datasets. For example, the AUC score of RG-Explainer on Tree-Grid is 0.927 while that of the runner-up is only 0.714, leading to an improvement of 29.8%. On the BA-Shapes dataset, RG-Explainer achieves comparable results with the winner's and significantly outperforms GNNExplainer. These results show the advantage of applying reinforcement learning techniques in constructing explanatory subgraphs. Further, since the constructed subgraphs are connected, they could better characterize motifs in the graph. Note that in the graph classification task, our method uses a locator to first select the seed nodes. We also test the performance of the locator and find that the locator selects $\sim 66\%$ and $\sim 84\%$ accurate seed nodes (i.e., nodes in the ground-truth motif) for BA-2Motifs and MUTAG, respectively. This further explains the good performance of RG-Explainer for graph classification.

## 5.4   Inductive setting

We further test the performance of RG-Explainer in the inductive setting. We compare it with PGExplainer, which are both learning-based methods. Specifically, we vary the training set sizes from $\{10\%, 30\%, 50\%, 70\%, 90\%\}$ and take the remaining instances for testing. For each dataset, we run the experiments 10 times and compute the average AUC scores.

Due to the limited space, Fig 2 only shows the results on BA-Community, Tree-Grid and MUTAG datasets. For other datasets, see the supplementary material. The figure also includes the results of both methods in the transductive setting (i.e., use all the instances) for reference. From the figure, RG-Explainer generally outperforms PGExplainer as the training set size increases. For example, with only 10% training instances in the Tree-Grid dataset, RG-Explainer significantly outperforms PGExplainer by a large margin. This shows that RG-Explainer generalizes better than PGExplainer.

## 6 Conclusion

We present RG-Explainer to generate the instance-level explanations for GNNs in this paper. The RL-based generator is proposed to ensure the message passing nature of GNNs. Besides, we design the seed locator and stopping criteria to find the most influential node in a graph instance and check whether the generated explanatory graph is good enough, respectively. Though our method increases the transparency of GNN predictions, it may put GNN models at a high risk of being attacked. How to utilize the GNN explanations to make GNN models more robust is a future research direction.

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
