# Supplementary: Reinforcement Learning Enhanced Explainer for Graph Neural Networks

**Caihua Shan**[1], **Yifei Shen**[2], **Yao Zhang**[3], **Xiang Li**[4], **Dongsheng Li**[1]
[1]Microsoft Research Asia
{caihua.shan,dongsheng.li}@microsoft.com
[2]The Hong Kong University of Science and Technology
yshenaw@connect.ust.hk
[3]Fudan University
yaozhang@fudan.edu.cn
[4]East China Normal University
xiangli@dase.ecnu.edu.cn

## A. Explanation algorithms

We show the pseudocode of our RG-Explainer for node classification and graph classification in Alg. 1 and 2, respectively.

### A1. Node Classification

Given an input graph $G = (\mathcal{V}, \mathcal{E})$ and its features $\mathcal{X}$, a trained GNN model $f$ and node instances $\mathcal{I}$, the goal is to train our graph generator $\mathbb{G}$ and find the explanatory subgraph for node instances.

We first extract the computation graph $G^{(i)}$ for each node $v_i \in \mathcal{I}$ (line 4). The reason is that the prediction of a node instance is decided by its $L$-hop neighborhoods because of the message passing scheme. $L$ is the number of GNN layers in the trained model $f$. Here $L$ is 3. The goal of pre-training is to let $\mathbb{G}$ know how to generate a subgraph from a node sequentially (line 5-7). Therefore, we random sample some valid layer-wise generation sequences $D^{(i)}$ for $G^{(i)}$ as pre-training samples. These sequences $D^{(i)}$ are also truncated with random lengths to let $\mathbb{G}$ generate the subgraph with different sizes.

In each training epoch, we generate $S^{(i)}$ for each node $v_i \in \mathcal{I}$ by $\mathbb{G}$ (line 10-19). We compute the prediction loss and regularizations as (negative) rewards for the set of current explanatory subgraphs $\{S^{(i)}\}$ and update the parameters in $\mathbb{G}$ (line 20-21). In the inference phase, we infer the explanatory subgraph $S_{\text{final}}^{(i)}$ for node instances with trained graph generator $\mathbb{G}$ (line 23-24).

### A2. Graph Classification

We show our RG-Explainer for graph classification in Alg. 2. The algorithm is similar to the one explaining node classifications, except that we train our seed locator to detect the most influential node in a graph instance. Given a set of graph instances $G^{(n)} \in \mathcal{I}$ and a trained GNN model $f$, the goal is to obtain the seed locator $\mathbb{L}$, graph generator $\mathbb{G}$ and the explanatory subgraph $S_{\text{final}}^{(n)}$.

We first pretrain $\mathbb{L}$. The final graph representations $z_{g_n}$ and node representations $z_{v_{i,n}}$ can be computed by the trained GNN model $f(\cdot)$ (line 4). They are the input of $\mathbb{L}$. We then build the pre-training data $D$ by obtaining the computation graph of each node in the graph instance, and computing its corresponding reward (line 5-6). We use $D$ to let $\mathbb{L}$ learn which kind of seed nodes have the highest reward. Given $\mathbb{L}$, we pretrain $\mathbb{G}$ similar to what we do for node classifications (line 10-13).

35th Conference on Neural Information Processing Systems (NeurIPS 2021), Sydney, Australia.

---

**Algorithm 1** Training algorithm for explaining node classification

---

1: **Input:** The input graph $G = (\mathcal{V}, \mathcal{E})$, node features $\mathcal{X}$, node instances $\mathcal{I}$, and a trained GNN model $f(\cdot)$.
2: **Output**: Our graph generator $\mathbb{G}$ and explanatory subgraphs for nodes in $\mathcal{I}$.

3: *# Pre-train*
4: $G^{(i)} \leftarrow$ the computation graph for node $v_i \in \mathcal{I}$.
5: $D^{(i)} \leftarrow$ a sampled valid layer-wise sequence to generate $G^{(i)}$ truncated with random length.
6: Pre-train $\mathbb{G}$ on $\{D^{(i)}\}$ by optimizing the list-wise MLE.
7: Pre-train $\mathbb{G}$ on $\{D^{(i)}\}$ by optimizing the set-wise MLE.

8: *# Training phase*
9: **for** each training epoch **do**
10:     **for** each node $v_i \in \mathcal{I}$ **do**
11:         Initialize $S_0^{(i)} = \{v_i\}$.
12:         **for** each timestamp $t = \{1...T\}$ **do**
13:             $X'_t \leftarrow$ augmented node features in Eq. 1.
14:             $\bar{H}_t^L \leftarrow$ node representation propagated by APPNP and transformed by MLP in Eq. 2 and Eq. 3.
15:             Update $S_t^{(i)}$ by adding a node $v_t$ based on Eq. 4.
16:             Check the stopping criteria by Eq. 10.
17:         **end for**
18:         $S^{(i)} \leftarrow$ current generated explanatory subgraph by $\mathbb{G}$ for node $v_i \in \mathcal{I}$.
19:     **end for**
20:     $\mathcal{L}(\{S^{(i)}\}_{v_i \in \mathcal{I}}) \leftarrow$ computed prediction loss and regularizations as reward with Eq. 7.
21:     Update parameters $\{\Theta_1, \Theta_2, \theta_3, \theta_4\}$ in $\mathbb{G}$ with policy gradient by Eq. 8 and Eq.9.
22: **end for**

23: *# Inference phase*
24: $S_{\text{final}}^{(i)} \leftarrow$ generated explanatory subgraph by $\mathbb{G}$ for node $v_i \in \mathcal{I}$.

---

---

**Algorithm 2** Training algorithm for explaining graph classification

---

1: **Input:** Graph instances $G^{(n)} \in \mathcal{I}$, and a trained GNN model $f(\cdot)$.
2: **Output**: Our seed locator $\mathbb{L}$, graph generator $\mathbb{G}$ and explanatory subgraphs for graphs in $\mathcal{I}$.

3: *# Pre-train $\mathbb{L}$*
4: $z_{g_n}/z_{v_{i,n}} \leftarrow$ the graph/node representation produced by $f(G^{(n)})$.
5: **for** each sampled graph $G^{(n)} \in \mathcal{I}$ **do**
6:     $D = [C_{v_i}, -\mathcal{L}(C_{v_i})] \leftarrow$ the training pair of computation graph of node $v_i \in G^{(n)}$ and its reward.
7:     Pre-train parameters in $\mathbb{L}$ on $D$.
8: **end for**

9: *# Pre-train $\mathbb{G}$*
10: $\widetilde{G}^{(n)} \leftarrow$ the computation graph of selected node $\mathbb{L}(G^{(n)})$ for graph $G^{(n)} \in \mathcal{I}$.
11: $D^{(n)} \leftarrow$ a sampled valid layer-wise sequence to generate $\widetilde{G}^{(n)}$ truncated with random length.
12: Pre-train $\mathbb{G}$ on $\{D^{(n)}\}$ by optimizing the list-wise MLE.
13: Pre-train $\mathbb{G}$ on $\{D^{(n)}\}$ by optimizing the set-wise MLE.

14: *# Training phase*
15: **for** each training epoch **do**
16:     *# Coordinate train $\mathbb{G}$*
17:     **for** each graph $G^{(n)} \in \mathcal{I}$ **do**
18:         Initialize $S_0^{(n)} = \{\mathbb{L}(G^{(n)})\}$.
19:         $S^{(n)} \leftarrow$ current generated explanatory subgraph by $\mathbb{G}$ step by step.
20:     **end for**
21:     $\mathcal{L}(\{S^{(n)}\}) \leftarrow$ computed prediction loss and regularizations as reward with Eq. 11.
22:     Update parameters $\{\Theta_1, \Theta_2, \theta_3, \theta_4\}$ in $\mathbb{G}$ with policy gradient by Eq. 8 and Eq. 9.
23:     *# Coordinate train $\mathbb{L}$*
24:     **for** each sampled graph $G^{(n)} \in \mathcal{I}$ **do**
25:         $D = [\mathbb{G}(v_i), -\mathcal{L}(\mathbb{G}(v_i))] \leftarrow$ the training pair of generated graph of node $v_i \in G^{(n)}$ and its reward.
26:         Update parameters in $\mathbb{L}$ on $D$.
27:     **end for**
28: **end for**

29: *# Inference phase*
30: $S_{\text{final}}^{(n)} \leftarrow$ generated explanatory subgraph by $\mathbb{G}(\mathbb{L}(G^{(n)}))$ for graph $G^{(n)} \in \mathcal{I}$.

---

In each training epoch, we coordinately train $\mathbb{L}$ and $\mathbb{G}$. Given a fixed $\mathbb{L}$, the steps to train $\mathbb{G}$ are the same as Alg. 1 (line 17-22). Given a fixed $\mathbb{G}$, we sample graph instances, and then build training data by generating the subgraph and computing the reward from each node in sampled graph instance. Based on these training samples, we update the parameters in $\mathbb{L}$. In the inference phase, we infer the explanatory subgraph $S_{\text{final}}^{(n)}$ for graphs instances with trained seed locator $\mathbb{L}$ and graph generator $\mathbb{G}$ (line 29-34).

## B. Implementation Details

All experiments are conducted on a Linux machine with an NVIDIA Tesla P100 GPU with 10.2 CUDA. RG-Explainer is implemented with Python 3.7 and Pytorch 1.8.0. The hyper-parameters in RG-Explainer are listed in the following table. We also releases the code in the supplementary.

Table 1: Hyper-parameters in RG-Explainer

| Hyper-parameters | Value |
|---|---|
| Number of pre-training epochs for list-wise MLE | 10 |
| Number of pre-training epochs for set-wise MLE | 25 |
| Sample ratio of graph instance to pre-train $\mathbb{L}$ | 1.0 |
| Batch size to pre-train $\mathbb{G}$ | 32 |
| Batch size to train $\mathbb{G}$ | 128 |
| Number of layers of APPNP in $\mathbb{G}$ | 3 |
| $\alpha$ in APPNP in $\mathbb{G}$ | 0.85 |
| Hidden dimension in $\mathbb{G}$ | 64 |
| Architecture of MLP in $\mathbb{L}$ | 64-8-1 |
| Learning rate | 1e-2 |
| Optimizer | Adam |
| Number of rollouts | 5 |
| Number of hops | 3 |
| Maximum size of generated sequences | 20 |
| Training epochs (node tasks) | Searched from {30, 50} |
| Training epochs (graph tasks) | 10 |
| Sample ratio of graph instance to train $\mathbb{L}$ | 0.2 |
| Coefficient of size loss | 0.01 |
| Coefficient of similarity loss (node tasks) | 1.0 |
| Coefficient of similarity loss (graph tasks) | 0.01 |
| Coefficient of radius penalty | Searched from {0.1, 0.01, 0.0} |

## C. Potential Negative Societal Impacts

Our RG-explainer increases the transparency of GNN models, i.e., we know the reason for GNN predictions. It may put GNN models at a high risk of being attacked. Some algorithms for graph attacks could utilize our method to obtain the most influential subgraph and perturb this subgraph. It is a serious alert to technology companies who maintain the platforms and operate various applications based on GNN algorithms. However, we believe that the graph defense approaches are also beneficial from our method to protect the most influential subgraph and make the predictions robust.

## D. Inductive Setting

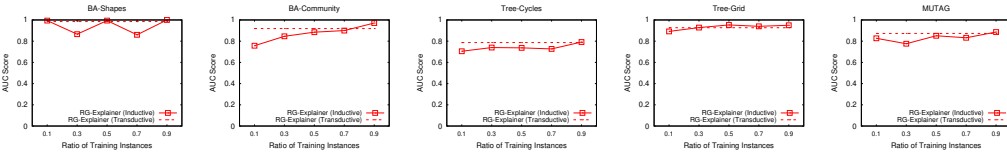

Figure 1: RG-Explainer in the inductive setting.

In this part, we show the performance of RG-explainer in the inductive setting. Specifically, we vary the training set sizes from {10%, 30%, 50%, 70%, 90%} and take the remaining instances for testing. With the increase of training samples, the AUC scores are steadily rising (e.g., BA-Community, Tree-Grid and MUTAG). For BA-Shapes and Tree-Cycles, they already have enough training samples when the ratio is 10%. Thus, their performances fall in a certain interval.

## E. Convergence Analysis

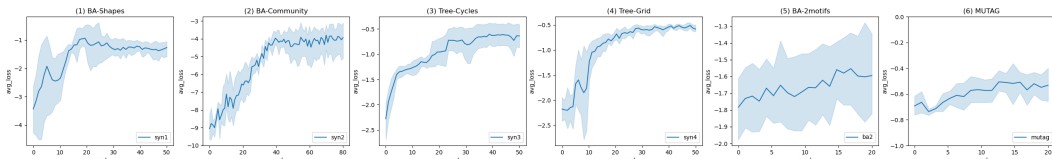

Figure 2: Convergence Analysis

In Figure 2, we experimentally observe that RG-Explainer could converge on all the datasets, where the x-axis represents the training epoch and the y-axis indicates the negative loss value (reward).

## F. Ablation Study

Table 2: Ablation Study

|  | Node Classification | | | | Graph Classification | |
|---|---|---|---|---|---|---|
|  | BA-Shapes | BA-Community | Tree-Cycles | Tree-Grid | BA-2motifs | MUTAG |
| w/o size loss | 0.958 | 0.871 | 0.851 | 0.880 | 0.592 | 0.820 |
| w/o similarity loss | 0.965 | 0.832 | 0.730 | 0.763 | 0.608 | 0.856 |
| w/o radius penalty | 0.991 | 0.885 | 0.821 | 0.927 | 0.566 | 0.861 |
| w/o pre-training strategy | 0.986 | 0.707 | 0.5 | 0.5 | 0.547 | 0.683 |

In this section, we first analyze the effect of regularization terms. We set the coefficient of a regularization term as 0 to remove its effect. The result is shown in Table 2. Generally speaking, the effects of size loss and similarity loss are more important. For example, the AUC scores of BA-Shapes and MUTAG decrease by 3% and 6% without the size loss. The AUC scores of BA-Community and Tree-Grid decline by 9% and 18% without the similarity loss. Because size loss and radius penalty work together to control the size of generated explanatory subgraph, the impact of removing the radius penalty is small.

We further study the importance of our proposed pre-training strategy. The results are shown in the bottom line of the table. For the easy case (BA-Shapes), RG-Explainer can learn the accurate explanatory graph. However, for other datasets, the performance of RG-Explainer degenerates significantly. This shows the importance of the pre-training step.