# OpenReview forum: "Reinforcement Learning Enhanced Explainer for Graph Neural Networks"
_NeurIPS.cc/2021/Conference — NeurIPS 2021 Poster_

### Official Review · Reviewer_MN97 · 2021-07-16

**Rating:** 6
**Confidence:** 4

**Summary:**

This paper proposes an RL-based GNN explainer to address the neglected interactions among nodes and edges in the generated subgraph for model interpretation in the existing works. Specifically, the subgraph generation process is modeled as a Markov decision process (MDP) where the state is the currently selected node, action space is composed by the candidate neighboring nodes, and reward signal is the combinatorial loss to encourage the RL learner to generate a compact and succinct subgraph. In this way, the subgraph generation process strictly follows the structure of the original graph and therefore considers the interactions among nodes and edges.

**Main Review:**

Strengths:

1. In general, this paper is well-written and easy to follow.
2. The proposed model is technically sound. Specifically, the definition of MDP seems reasonable, the reward signal is properly designed and well-motivated and the sparse reward signal problem is addressed properly with Monte-Carlo estimation. In addition, stopping criteria are designed to avoid large subgraphs with a maximum number of generation steps; and the starting point selection with the learning-based approach is designed to address graph classification scenarios efficiently.
3. The qualitative evaluation and quantitative results show the superior outcome comparing with state-of-the-art approaches.

Weaknesses:
1. Policy gradient and Monte-Carlo estimation may suffer from high variances and could be unstable, which may not be able to generate explanation consistently. Monte-Carlo estimation could also be inefficient. More analysis on rollout reward evolution with respect to the training timestamps may help.
2. Line 200 "In this way, we can solve the sparse reward problem and also distribute reward signals at all steps." The sparse reward is a very challenging problem. I understand that Monte-Carlo estimation could help since it gives a reward to each step. However, Monte-Carlo estimation suffers from high variance, which could slow down the convergence. To my knowledge, this is far from "solve " sparse rewards. A term like "alleviate" could be more appropriate. I would also expect more discussion of why Valina Policy Gradient and Monte-Carlo estimation are adopted here, why they can tackle the sparse reward, and why not using other RL algorithms that could be more stable and may generate more consistent explanations, like DQN or A2C.
3.More justification of leveraging the self-attention mechanism to generate stop signal is needed as it seems a bit arbitrary to directly incorporate the \tilde{H}_t(STOP) into action space. In addition, the author mentioned that a maximum number of generation steps is still needed to avoid generating large subgraphs. It is unclear whether the self-attention mechanism is effective. Ablation studies with respect to the sizes of the generated graph without maximum generation steps would be helpful.

-------------------------------- Post rebuttal --------------------------------

The authors' response has addressed my concerns. The score is raised.



**Time Spent Reviewing:**

5

---

> ### Author Response · Authors · 2021-08-10
> **To Review MN97**
>
> We sincerely thank the reviewer for the helpful comments and we next answer all the raised questions.
>
>
> **Q1:** Policy gradient and Monte-Carlo estimation may suffer from high variances and could be unstable. Why do we use Valina policy gradient and monte-carlo estimation, but not DQN and A2C?
>
> **A1:** The main concern in w1 and w2 is whether Valina policy gradient and monte-carlo estimation are good enough to generate good explanations. In our problem setting, we argue that the variance of rollout rewards could be relatively small because
> 1. Our mdp is deterministic. Given a state (current subgraph) and an action (picked node), the next state (next subgraph) is deterministic.
>
> 2. The action space is limited by several factors:
> a. graph structure: each time we can only pick up a node from the boundary of the current graph;
> b. the maximum number of nodes: the size of the final explanatory graph is limited to a maximum value;
> c. the pre-training strategy: the pre-training strategies can be used to further shrink the space for node selection.
>
> Thus, RG-Explanier still performs better than baselines in our experiments.
>
> For DQN and A2C, they utilize a function/neural network to approximate V-value, Q-value or advantage function. The function approximation is biased estimation with low variance; while monte-carlo estimation is unbiased estimation with high variance. Since we believe the variance of rollout rewards is relatively small, monte-carlo estimation has its advantages. But we also agree that DQN and A2C can be tried in the future work.
>
> Meanwhile, we also conduct the convergence analysis on RG-Explainer. We experimentally observe RG-Explainer converges in all the datasets. The results are given in https://www.dropbox.com/s/j94mywra3dlb5zw/loss.png?dl=0, where the x-axis represents the training epoch and the y-axis indicates the negative loss value (reward). It further supports our argument above.
>
>
>
>
> **Q2:** Sparse reward problem: "alleviate" is more appropriate
>
> **A2:** We agree that "alleviate" is more appropriate. We have revised our paper accordingly.
>
>
> **Q3:** Effectiveness of the self-attention mechanism
>
> **A3:** The reason for the self-attention mechanism to generate stop signals is that the self-attention mechanism can be used to aggregate the information in the current explanatory graph after message passing and judge whether to stop.
>
> We further conduct ablation studies by deleting the maximum generation steps limitation to show the importance of the self-attention mechanism. For fairness, we compare the training time per epoch, AUC and sizes of generated graph per epoch using the same seed. The results are given as follows. We see that, without maximum generation steps limitation, RG-Explainer will generate larger subgraphs, which causes longer training time but the AUC scores are comparable. These results demonstrate the effectiveness of the self-attention mechanism.
>
>
> | Training time per epoch        |  BA-Shapes   | BA-Community  |  Tree-Cycles   | Tree-Grid  |  BA-2motifs   |  MUTAG   |
> |:----:                    |  :----:      |:----:         | :----:         |:----:      | :----:        | :----:   |
> | RG-Explainer                 |  218.2s      | 180.7s         | 129.8s         |  204.5s    | 85.6s         | 210.4s   |
> | w/o maximum  generation steps|  561.5s      | 255.6s         | 177.1s         |  765.3s     |   176.8s         |  343.4s    |
>
>
> | AUC          |  BA-Shapes   | BA-Community  |  Tree-Cycles   | Tree-Grid  |  BA-2motifs   |  MUTAG   |
> |:----:             |  :----: |:----: | :----: |:----:  | :----: | :----:|
> |RG-Explainer                       | 0.992   | 0.883 | 0.883  | 0.927  | 0.657  | 0.873 |
> |w/o maximum  generation steps      | 0.990   | 0.837 | 0.866  | 0.915  | 0.689  | 0.791 |
>
>
> | Sizes of generated graph per epoch    |  BA-Shapes   | BA-Community  |  Tree-Cycles   | Tree-Grid  |  BA-2motifs   |  MUTAG   |
> |:----:                        |  :----:      |:----:         | :----:         |:----:      | :----:        | :----:   |
> | RG-Explainer                 |  18.24      |  13.98         | 13.03         |  17.6    |  8.8      | 9.7   |
> | w/o maximum  generation steps|  17.11      | 14.56          | 13.76         |   24.3     |  14.2          | 12.2    |

---

> > ### Comment · Reviewer_MN97 · 2021-08-21
> > **Thanks for the response**
> >
> > The authors' response has addressed my concerns. I will raise my score.

---

### Official Review · Reviewer_GKcV · 2021-07-17

**Rating:** 5
**Confidence:** 3

**Summary:**

This paper studied the explanation for GNNs. The authors formulated the task of generating an exploratory sub-graph as a combinatorial optimization problem, and proposed a framework, RG-Explainer, to solve the problem via reinforcement learning.

The RG-Explainer framework consists of three parts: a seed locator for starting node selection, a graph generator for generating the explanatory sub-graph, and a learned stopping criteria to avoid generating very large explanatory graphs.

The authors conducted experiments on node classification and graph classification tasks.  The qualitative results showed that the proposed method could provide plausible explanations. The quantitative results showed that the proposed method has better accuracy, and outperforms the baseline in both inductive and transductive settings.


**Limitations And Societal Impact:**

Limitations have been discussed before, and there's no obvious negative societal impact.

**Main Review:**

Originality: The proposed framework is novel in terms of GNN explanatory. The authors introduced RL-based learning schema for the explanatory sub-graph generation.

Quality: The proposed framework is reasonable. My main concerns are about the experiment results.
1)	It seems that the authors did not implement the baselines based on the officially released code (e.g., https://github.com/flyingdoog/PGExplainer for [16]), but the reproduction challenge ([11]), which reports much lower results compared to the original paper.  A proper justification will be appreciated.
2)	More ablation studies are necessary to clarify the contribution of different components (e.g., the different components in Eq. (7), and the pre-training part).
3)	Whether this RL-based framework requires longer training / inference time?The efficiency analysis, as discussed in existing works [16, 34], would be good to be included.

Clarity: Overall, this paper is written clearly, except that Figure 1 is a bit confusing. In particular,
1)	Does the locator receive the reward from the environment?
2)	In the graph generator, should the reward arrows within the episode be removed, as the reward is assigned episodically?
3)	What is the meaning of the “right” ticks?

Significance: The significance is moderate. Although RL has been extensively studied in solving combinatorial optimization problems (e.g., the routing problems), this paper provides a new direction for RL application.




**Time Spent Reviewing:**

5

---

> ### Author Response · Authors · 2021-08-10
> **To Review GKcv**
>
> We sincerely appreciate the anonymous reviewer for the insightful comments. Here are our responses to the questions raised.
>
>
> **Q1:** Justification that we utilize the reproduction version [11] instead of the officially released version for PGExplainer [16].
>
> **A1:** There are two reasons why we use [11] but not [16].
> On the one hand, [11] provides a high-quality code implementation and a fair benchmark for baselines.
> On the other hand, [11] also fixes some implementation issues in the original codebases.
> Specifically,
>
> a. The officially released version [16] is a Tensorflow version which only includes the implementation of PGExplainer. Reproduction version [11] is one of Pytorch implementations linked in the [16]'s GitHub (https://github.com/flyingdoog/PGExplainer). Further, [11] has been formally peer-reviewed, discussed with original authors and accepted by ML Reproducibility Challenge 2020. [11] not only implements GNNExplainer and PGExplainer, but also tracks experimental configurations to ensure that all experiments can be replicated. Therefore, we argue that [11] is a high-quality code implementation and a fair benchmark for comparison.
>
> b. The reproduction [11] reports a lower AUC score compared to the original paper because it utilizes slightly different GCN layers and batch normalizations (please see https://openreview.net/forum?id=8JHrucviUf&noteId=FdK9s0fkXib). The original version [16] uses the GCNConv, where each layer is represented by $f(H^{(l)},A)=\sigma(W^{(l)}AH^{(l)})$. The reproduction [11] adopts GCNConv from torch_geometric, which is $f(H^{(l)},A)=\sigma(AH^{(l)}W^{(l)})$. Actually, the latter one is a more general implementation for GCN (please see Equation 2 in the GCN paper https://openreview.net/pdf?id=SJU4ayYgl). The latter one performs well both in the transductive and inductive settings, while the former one performs poorly in the inductive setting. For the batch normalization, the original version [16] contains an error that results in the batch-normalization layers being kept in training mode during evaluation. However, the reproduction [11] fixes this issue. The different GCN layers and batch normalizations in the trained model lead to different AUC scores for GNNExplainer and PGExplainer in [11] and [16], especially in graph classification tasks.
>
> **Q2:** Ablation studies about different components in Eq. (7) and the pre-training part.
>
> **A2:** We strongly agree that ablation studies are needed to show the contribution of different components in RG-explainer. In the original manuscript, we have included the ablation study in Section E of the supplementary material, which tests the effects of components in Eq.7 (size loss, similarity loss and radius penalty). The results are shown in the following table. From the table, we see that the effects of size loss and similarity loss are more important.
>
> | AUC          |  BA-Shapes   | BA-Community  |  Tree-Cycles   | Tree-Grid  |  BA-2motifs   |  MUTAG   |
> |:----:             |  :----: |:----: | :----: |:----:  | :----: | :----:|
> |RG-Explainer       | 0.985   | 0.919 |  0.787 | 0.927  | 0.657  | 0.873 |
> |w/o size loss      | 0.958   | 0.871 | 0.851  |  0.880 | 0.592  | 0.820 |
> |w/o similarity loss| 0.965   | 0.832 | 0.730  |  0.763 | 0.608  | 0.856 |
> |w/o radius penalty | 0.991   | 0.885 | 0.821  |  0.927 | 0.566  | 0.861 |
> |w/o pre-training strategy      | 0.986   | 0.707 | 0.5  |  0.5 | 0.547  | 0.683 |
>
> We further study the importance of our proposed pre-training strategy. The results are shown in the bottom line of the table. For the easy case (BA-Shapes), RG-Explainer can learn the accurate explanatory graph. However, for other datasets, the performance of RG-Explainer degenerates significantly. This further shows the importance of the pre-training step.
>
> **Q3:** Efficiency analysis
>
> **A3:** We follow the same setting in [16] to compare the inference time of GNNExplainer, PGExplainer and RG-Explainer with an NVIDIA Tesla P100 GPU. The results are shown in the following table. On the one hand, PGExplainer and RG-Explainer are learning-based methods, which have great generalization ability and can be utilized in an inductive setting (demonstrated in section 5.4). The time to explain a new instance is the inference time of the trained models. We see that the inference time of RG-Explainer and PGExplainer are in the same order of magnitude. On the other hand, GNNExplainer and other searched-based methods (e.g., [34]) need to retrain the model to explain a new instance. The inference time of GNNExplainer is 10x slower than that of RG-Explainer, and it has been reported in [34] that the time of other searched-based methods is larger than 1s.
>
>
> | Inference time    |  BA-Shapes   | BA-Community  |  Tree-Cycles   | Tree-Grid  |  BA-2motifs    |  MUTAG   |
> |:----:             |  :----:      |:----:         | :----:         |:----:       | :----:        | :----:   |
> |GNNExplainer       |  736ms       | 1036ms        | 899ms          | 2814ms      | 361ms         | 178ms    |
> |PGExplainer        |  55ms        | 81ms          | 6ms            | 7ms         | 4ms           | 26ms     |
> |RG-Explainer       |  22ms        | 18ms          | 16ms           | 20ms          |  10ms         | 12ms         |
>
>
> We also show the training time of our method RG-explainer including the pre-training part and iterative update per epoch. The total training time on all the datasets is about 1\~3 hrs. Based on the two tables, we can conclude that learning-based methods (e.g. RG-Explainer) are more practical for large-scale datasets. For example, the learning-based methods could train an explanatory model with $10^3$ instances in 1\~3 hrs and explain $10^6$ instances (a large-scale dataset) in $10^6$*100ms \~ 2.8 hrs. However, searched-based methods will spend $10^6$*1s ~ 11 days to explain all the instances.
>
>
> |   Training time    |  BA-Shapes   | BA-Community  |  Tree-Cycles   | Tree-Grid  |  BA-2motifs   |  MUTAG   |
> | :----:                    |  :----:      |:----:         | :----:         |:----:      | :----:        | :----:   |
> |pre-training              |  213.9s      | 281.6s        | 193.8s         |  1013.0s   | 2643.7s       | 4435.5s  |
> |iterative update per epoch|  218.2s      | 180.7s        | 129.8s         |  204.5s    | 85.6s         | 210.4s   |
> | default epochs           |  50          | 50            | 30             |  30        | 10            | 10       |
> | total                    |  3hrs        | 2.5hrs        | 1.1hrs         |  2hrs   | 1 hrs         | 1.8hrs   |
>
> **Q4:** Clarity for Figure 1
>
> **A4:** Figure 1 is the framework of our RG-explainer, which contains two main components: a locator and a generator. The locator selects a seed node and feeds it into the generator. Based on the seed node, the generator then generates a complete explanatory graph and obtains the final reward from the environment. This reward is used as feedback to update the generator and the locator. We alternatively optimize the generator and locator until convergence.
>
> To alleviate the confusion of Figure 1, we next reply to each question raised.
>
> Q4.1 Does the locator receive the reward from the environment?
>
> A4.1 Yes, the locator receives the reward through the generator from the environment and it is trained based on this reward.
>
> Q4.2 In the graph generator, should the reward arrows within the episode be removed, as the reward is assigned episodically?
>
> A4.2 Yes, the generator only revieces the final reward after generating a complete explanatory graph (i.e., when an episode ends). The wavy arrows in the episode mean rollout rewards estimated by the monte-carlo estimation. We agree with your point and we have removed the reward arrows in our revised paper.
>
> Q4.3 What is the meaning of the “right” ticks?
>
> A4.3 The 'right' ticks show that we select a node in each step and add it to the explanatory graph. After we select the "STOP" node, the generation process ends and a complete explanatory graph is generated.

---

### Official Review · Reviewer_gLix · 2021-07-17

**Rating:** 6
**Confidence:** 3

**Summary:**

This paper proposes an RL-enhanced GNN explainer, RG-Explainer which could construct a connected explanatory subgraph by sequentially adding nodes from the boundary of the current generated graph, and can be used for inductive setting. Experiments on both synthetic and real datasets are carried out to compare the proposed algorithm with existing  GNN explainers.

**Limitations And Societal Impact:**

Yes, but the discussion of societal impact is not in the main text but in the supplementary document.

**Main Review:**

The reviewer finds this manuscript well-written and the performance of the algorithm impressive. The reviewer believes such a framework could be useful for understanding and explain the performance of GNNs.

One major question is regarding the complex convergence of the algorithm. Due to the introduction of the RL framework, which leads to high training complexity, and the policy gradient may not necessarily converge. However, no analysis or experiments were provided regarding this issue. More discussion with respect to the convergence of the framework is needed.

**Time Spent Reviewing:**

3

---

> ### Author Response · Authors · 2021-08-10
> **To Review gLix**
>
> We sincerely thank the anonymous reviewer for the constructive comments. Here are our responses to the reviewer's questions.
>
> **Q1:** Convergence of the proposed algorithm
>
> **A1:** For the convergence of our method RG-Explainer, we experimentally observe that
> RG-Explainer could converge on all the datasets.
> Please see the figure in https://www.dropbox.com/s/j94mywra3dlb5zw/loss.png?dl=0,
> where the x-axis represents the training epoch and the y-axis indicates the negative loss value (reward).
> We have included the figure in the revised paper.
>
> We also analyze the reasons why Valina policy gradient and monte-carlo estimation perform well in our experiments. Please see the answer A1 of Reviewer 3.

---

### Decision · Program_Chairs · 2021-09-27

**Decision:**

Accept (Poster)

**Comment:**

The paper addresses the issue of explainability in GNNs using RL for finding explanatory subgraphs. The authors provided an extensive rebuttal including new experimental results requested by the reviewers. The AC believes the authors' responses address in a satisfactory manner the reviewers' comments and recommends acceptance.